# Efficient Potential-based Exploration in Reinforcement Learning using Inverse Dynamic Bisimulation Metric

**Yiming Wang[1]**    **Ming Yang[1]**    **Renzhi Dong[1]**    **Binbin Sun[1]**    **Furui Liu[2]**

**Leong Hou U[1]***

[1]State Key Laboratory of Internet of Things for Smart City, University of Macau, Macao SAR, China
[2]Zhejiang Lab, Hangzhou, China
`{wang.yiming,ming.ink,renzhi.dong,sun.binbin}@connect.um.edu.mo`
`liufurui@zhejianglab.com,ryanlhu@um.edu.mo`

## Abstract

Reward shaping is an effective technique for integrating domain knowledge into reinforcement learning (RL). However, traditional approaches like potential-based reward shaping totally rely on manually designing shaping reward functions, which significantly restricts exploration efficiency and introduces human cognitive biases. While a number of RL methods have been proposed to boost exploration by designing an intrinsic reward signal as exploration bonus. Nevertheless, these methods heavily rely on the count-based episodic term in their exploration bonus which falls short in scalability. To address these limitations, we propose a general end-to-end potential-based exploration bonus for deep RL via potentials of state discrepancy, which motivates the agent to discover novel states and provides them with denser rewards without manual intervention. Specifically, we measure the novelty of adjacent states by calculating their distance using the bisimulation metric-based potential function, which enhances agent exploration and ensures policy invariance. In addition, we offer a theoretical guarantee on our inverse dynamic bisimulation metric, bounding the value difference and ensuring that the agent explores states with higher TD error, thus significantly improving training efficiency. The proposed approach is named **LIBERTY** (exp**L**oration v**I**a **B**isimulation m**EtR**ic-based s**T**ate discrepanc**Y**) which is comprehensively evaluated on the MuJoCo and the Arcade Learning Environments. Extensive experiments have verified the superiority and scalability of our algorithm compared with other competitive methods.

## 1   Introduction

Reward shaping is a common method of transforming possible domain knowledge to redesign the reward function so that it guides the agent to explore state-action space more effectively. The potential-based reward shaping (PBRS) method Ng et al. [1999] is the first to demonstrate that policy invariance can be ensured if the shaping reward function takes the form of the difference between potential values. Existing reward shaping approaches, such as PBRS and its variants Devlin and Kudenko [2012], Harutyunyan et al. [2015], Li et al. [2023], mainly concentrate on generating additional rewards using potential values. However, they often assume that the shaping rewards derived from prior knowledge are entirely beneficial without considering their potential limitations. Moreover, the conversion of human prior knowledge into numerical values unavoidably requires human intervention, leading to subjective judgments and potential cognitive biases. The heavy reliance on human prior

---

*Corresponding author.

37th Conference on Neural Information Processing Systems (NeurIPS 2023).

knowledge presents a significant limitation in terms of scalability. More recently, exploration has been extensively investigated in the realm of deep RL, and a lot of empirically successful methods Raileanu and Rocktäschel [2020], Badia et al. [2019], Zhang et al. [2021] have been proposed. These methods rely on exploration bonuses that are generated intrinsically, which reward the agent for visiting states that are considered novel according to a certain measure, like the likelihood of a state under a learned density model, the error of a forward dynamics model, etc. These approaches have demonstrated their effectiveness in tackling challenging exploration problems. However, these intrinsic methods are either difficult to explain or only specific to some tasks, that even minor changes to the environment can lead to substantial degradation in performance. These methods heavily depend on the count-based episodic term in their exploration bonus, which becomes ineffective when each state is unique and cannot be counted. Additionally, policy variance may arise due to the failure of intrinsic reward generated by these methods to converge, which could cause the optimal policy of the original Markov Decision Process (MDP) to shift.

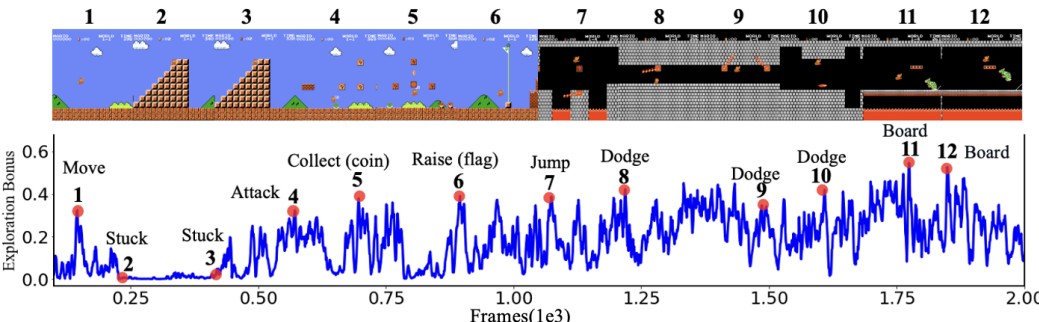

Figure 1: Illustration of LIBERTY rewards over the episodes of different stages in SuperMarioBros. The red points annotate the key frames. Many spikes are related to significant occurrences: moving forward (1), attacking enemies (4), collecting coins (5), raising the flag (6), jumping over obstacles (7), dodging higher level attacks (8,9,10), getting on the hoverboard (11,12). The reward is close to 0 when the agent is stuck (2,3).

A key idea in our work is to use a measure of discrepancy between states as the exploration bonus. Unlike exploration methods such as RIDE Raileanu and Rocktäschel [2020], which use the $\ell_2$ norm distance in the latent space to model state differences, we model the discrepancy between states based on their distance under the bisimulation metric Ferns et al. [2011]. Specifically, we propose a potential function based on the inverse dynamic bisimulation metric so that we can effectively explore the state space while ensuring that the learned optimal policy remains the same as the original MDP. Note that our method does not rely on any prior human knowledge, which sets it apart from other potential-based reward shaping techniques. In addition, we offer a theoretical guarantee on our inverse dynamic bisimulation metric, bounding the value difference and ensuring that the agent explores states with higher TD error, thus significantly improving training efficiency. As depicted in Figure 1, the exploration bonus is elevated in novel states (as indicated in the caption) across various stages of SuperMarioBros. This incentivizes the agent to explore actively, facilitating the acquisition of diverse "skills" throughout the learning process.

The main contributions of this paper are as follows. Firstly, we define an inverse dynamic bisimulation metric, serving as a potential function to ensure *policy invariance* without the need for any *prior human knowledge*. Secondly, we propose a general end-to-end exploration bonus for deep RL utilizing state discrepancy potentials. Compared to other exploration methods, our approach achieves more efficient exploration by encouraging agents to explore states with higher value difference (TD error), *without relying on count-based episodic terms*, which significantly improves the scalability of our approach. Lastly, extensive experiments are conducted in MuJoCo and the Arcade Learning Environments. The results demonstrate that our algorithm can effectively enhance exploration and accelerate training, while also confirming that our approach is highly scalable compared to other competitive methods.

## 2 Related Work

**Curiosity-driven Exploration.** Several exploration strategies Pathak et al. [2017], Burda et al. [2018], Houthooft et al. [2016], Pathak et al. [2019], Tao et al. [2020] use a dynamics model to generate curiosity to imporve exploration. Alternative approaches to modelling the environment's dynamics

are based on pseudo-counts Bellemare et al. [2016], Ostrovski et al. [2017], which use density estimations techniques to explore less seen areas of the environment. Some other studies Zheng et al. [2018], Hu et al. [2020] generate intrinsic rewards by neural networks to maximize the extrinsic return via meta gradient. There are also alternative methods that combine model-based intrinsic motivation with pseudo-counts. For example, RIDE Raileanu and Rocktäschel [2020] employs a reward mechanism that incentivizes the agent for transitions that have a significant impact on the state representation. NGU Badia et al. [2019] and NovelD Zhang et al. [2021] modulates a pseudo-count bonus with the intrinsic rewards provided by RND Burda et al. [2018]. It is worth noting that policy invariance from the original MDP could arise since the intrinsic reward of these methods is not guaranteed to converge.

**Potential-based Reward Shaping.** The first approach to guarantee policy invariance is potential-based reward shaping (PBRS) Ng et al. [1999]. This method defines the shaping reward function as the difference between values assessed through the potential function based on prior knowledge. There are numerous variants of PBRS, such as the potential-based advice (PBA) approach Wiewiora et al. [2003], which defines the potential function for providing advice on actions. Another variant is the dynamic PBRS approach Devlin and Kudenko [2012], which introduces a time parameter into potential function for allowing dynamic potentials. Additionally, the dynamic potential-based advice (DPBA) approach Harutyunyan et al. [2015] learns an auxiliary reward function for transforming any given rewards into potentials. More recent methods Gao and Toni [2015], Badnava et al. [2023], Grzes and Kudenko [2008] have shifted their focus to different areas within the field of reinforcement learning.

**Bisimulation Metric in RL.** Bisimulation relations Givan et al. [2003] group states into equivalence classes based on rewards and transition probabilities, but this method is prone to errors due to inaccurate estimates. Instead, Ferns et al. [2011, 2004], Ferns and Precup [2014] use a bisimulation metric that smoothly varies as rewards and transition probabilities change. Recently, Castro [2020] proposed an algorithm for on-policy bisimulation metrics. DBC Zhang et al. [2020] employs metric learning to approximate bisimulation-derived state aggregation. Goal-conditioned bisimulation Hansen-Estruch et al. [2022] captures functional equivariance, allowing for skill reuse in goal-conditioned RL. We provide a comprehensive comparison between our method and the other benchmarked methods in Appendix E.

## 3 Background

In this paper, we focus on the policy gradient framework Sutton et al. [1999] in the context of reinforcement learning (RL). We assume the underlying environment is a Markov decision process (MDP), defined by the tuple $\mathcal{M} = (\mathcal{S}, \mathcal{A}, \mathcal{P}, \mathcal{R}, \gamma)$, where $\mathcal{S}$ is the state space, $\mathcal{A}$ is the action space, $\mathcal{P}(s' \mid s, a)$ is state transition function from state $s \in \mathcal{S}$ to state $s' \in \mathcal{S}$, and $\gamma \in [0, 1)$ is the discount factor. Generally, the policy of an agent in an MDP is a mapping $\pi : \mathcal{S} \times \mathcal{A} \to [0, 1]$. An agent chooses actions $a \in \mathcal{A}$ according to a policy function $a \sim \pi(s)$, which updates the system state $s' \sim \mathcal{P}(s, a)$ yielding a reward $r = \mathcal{R}(s, a) \in \mathbb{R}$. In this paper, we denote a policy by $\pi_\theta$, where $\theta$ is the parameter of the policy function. The goal of the agent is to optimize the parameter $\theta$ for maximizing the expected accumulative rewards, $J(\pi_\theta) = \mathbb{E}_{\pi_\theta} \left[ \sum_{t=0}^{\infty} \gamma^t \mathcal{R}(s_t, a_t) \right]$.

**Potential-based Reward shaping.** Reward shaping refers to modifying the original reward function with a shaping reward function which incorporates domain knowledge. We consider the most general form, namely the additive form, of reward shaping. Formally, this can be defined as $\mathcal{R}'(s, a, s') = \mathcal{R}(s, a) + \mathcal{F}(s, a, s')$, where $\mathcal{R}(s, a)$ is the original reward function, $\mathcal{F}(s, a, s')$ is the shaping reward function, and $\mathcal{R}'(s, a, s')$ is the modified reward function. The original MDP tuple $\mathcal{M} = (\mathcal{S}, \mathcal{A}, \mathcal{P}, \mathcal{R}, \gamma)$ is transformed into the modified MDP tuple $\mathcal{M}' = (\mathcal{S}, \mathcal{A}, \mathcal{P}, \mathcal{R} + \mathcal{F}, \gamma)$. Early work of reward shaping Dorigo and Colombetti [1994] focuses on designing the shaping reward function $\mathcal{F}$, but ignores that the shaping rewards may change the optimal policy. While reward shaping can provide agents with useful feedback, it can also influence the optimal policy and lead to divergence if the reward function is not properly designed Snel and Whiteson [2012]. To address this problem, the Potential-based reward shaping (PBRS) function was introduced Ng et al. [1999]. PBRS reserves the optimality of policy if there exists a real-valued potential function $\Phi : \mathcal{S} \to \mathbb{R} \mid \forall (s, a, s') \in \mathcal{S} \times \mathcal{A} \times \mathcal{S}$, $\mathcal{F}$ is defined as the difference of potential values:

$$\mathcal{F}(s, a, s') = \gamma \Phi(s') - \Phi(s) \tag{1}$$

where $\gamma \in (0, 1]$ is the discount factor and $\Phi(s)$ is a potential function over all states.

**Bisimulation Metric.** Bisimulation is a technique for state abstraction that partitions different states $s_i$ and $s_j$ into groups that exhibit equivalent behavior Li et al. [2006]. A more compact definition has a recursive form: two states are bisimilar if they share both the same immediate reward and equivalent distributions over the next bisimilar states Givan et al. [2003] (definition in Appendix B). In continuous state spaces, finding exact partitions using bisimulation relations is typically impractical due to the high sensitivity of the relation to infinitesimal changes in the reward function or dynamics. For this reason, bisimulation metrics Ferns et al. [2011], Ferns and Precup [2014] softens the concept of state partitions, and instead defines a pseudometric space $(\mathcal{S}, d)$, where a distance function $d : \mathcal{S} \times \mathcal{S} \mapsto \mathbb{R}_{\geq 0}$ measures the "behavioral similarity" between two states. It is worth noting that $d$ is a pseudometric, allowing for a distance of zero between different states, indicating behavioral equivalence. However, the computational cost and the necessity of a tabular representation for states have limited the practicality of these methods for large-scale problems, such as continuous control. More recently, the on-policy bisimulation metric Castro [2020] (also called $\pi$-bisimulation) has been proposed as a solution to the aforementioned issue.

**Definition 1.** *(On-policy bisimulation metric Castro [2020]) Given a fixed policy $\pi$, the following on-policy bisimulation metric exists and is unique:*

$$d_\pi(s_i, s_j) = \left| r_i^\pi - r_j^\pi \right| + \gamma W_1(d_\pi)(\mathcal{P}^\pi(\cdot \mid s_i), \mathcal{P}^\pi(\cdot \mid s_j)) \tag{2}$$

*where $r_i^\pi = \mathbb{E}_{a \sim \pi}[\mathcal{R}(s_i, a)]$ and $\mathcal{P}^\pi(\cdot \mid s_i) = \mathbb{E}_{a \sim \pi}[\mathcal{P}(\cdot \mid s_i, a)]$.*

The above bisimulation metric, based on the Wasserstein metric $W_1$ Breugel and Worrell [2001], which is also known as the Earth Mover's Distance (EMD), is a meassure of how much the rewards collected in each state and the respective transition distributions differ. A distance of zero for a pair implies state aggregation, or *bisimilarity*.

## 4 Methodology

We propose **LIBERTY** (exp**L**oration v**I**a **B**isimulation m**E**t**R**ic-based s**T**ate discrepanc**Y**), utilizing potential-based exploration, which ensures both data efficiency and policy invariance. Bisimulation metrics are beneficial for state abstractions. However, prior methods have either trained distance functions specifically designed for the (fixed) policy evaluation setting Castro [2020], or utilized them for representation learning Zhang et al. [2020]. We are the first to propose a potential-based exploration framework that capitalizes on the discrepancy between consecutive states, as assessed by a bisimulation metric-based potential function. Furthermore, we present an inverse dynamic bisimulation metric designed to enhance effective exploration, which is proven to converge to a fixed point in Theorem 1.

**Intuition of Bisimulation Metric.** Several studies have utilized the $\ell_2$ norm distance to measure the difference between states for potential function evaluation or exploration bonus calculation. For instance, PBRS Ng et al. [1999] employs the $\ell_2$ norm distance to goal states as potential function, while RIDE Raileanu and Rocktäschel [2020] introduces a bonus that is calculated based on the $\ell_2$ norm distance between the embeddings of two consecutive states in the latent space. We argue that the $\ell_2$ norm is not well-suited for evaluating state differences as it does not consider the values of states. Therefore, in our work, we introduce the bisimulation metric as a more appropriate measure. In order to make a comparison with bisimulation metric used in our work, we project the states in SuperMarioBros onto a two dimensional latent space ($x$ axis and $y$ axis). Additionally, the $z$ axis represents the value of states, as illustrated in Figure 2. In our example, we have the

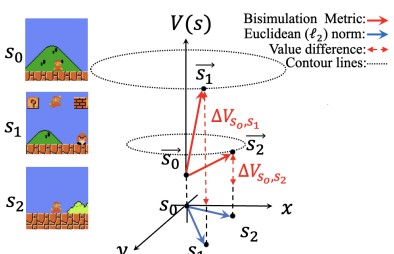

Figure 2: Metric comparison. $s(x, y)$: state $s$ projected into two dimensional latent space; $\overrightarrow{s}(x, y, z)$: state in three dimensional space where the $z$ axis denotes value $V(s)$.

following states: $s_0$ represents the initial state. $s_1$ corresponds to the state where Mario achieves the highest value by jumping to attack enemies. $s_2$ denotes the state with the second-highest value, where Mario simply moves forward. Given this scenario, the agent should receive a higher exploration

bonus to reach $s_1$ compared to $s_2$. As shown in Figure 2, the distance between $s_0$ and $s_1$ measured by the bisimulation metric is larger than the distance between $s_0$ and $s_2$. However, when considering the Euclidean ($\ell_2$ norm) distance, the distances are identical for $s_0$ to both $s_1$ and $s_2$. As a result, the agent utilizing the bisimulation metric prioritizes the exploration of novel states with higher TD error, which leads to significant improvements in policy training efficiency. In essence, the bisimulation metric $d$ provides a more accurate measure of distance between states compared to the $\ell_2$ norm distance used in RIDE and PBRS. We present the detailed theoretical analysis in section 5.

**Issues on Exploration via Bisimulation Metric.** Exploration using the bisimulation metric only may lead to meaningless exploration. Consider the 4th and 5th frame in Figure 1, an agent navigating a level in SuperMarioBros that features randomly moving monsters. In this case, the agent could potentially visit a vast number of different states and collect a large amount of cumulative bonus without taking any meaningful actions that promote exploration. Due to the frequent state changes, the state difference measured by the bisimulation metric remains high under this condition. We should identify the state difference caused by actions so that the exploration efficiency can be promoted. To avoid such meaningless exploration, we propose the inverse dynamic bisimulation metric.

**Inverse Dynamic Bisimulation Metric.** Given two consecutive observations, we train an inverse dynamic model Bromley et al. [1993], Koch et al. [2015] $I : \mathcal{S} \times \mathcal{S} \to \mathcal{A}$, which predicts the action $a_t \in \mathcal{A}$ that changed $\mathbf{s}_t$ to $\mathbf{s}_{t+1}$. The parameters of the inverse dynamic model $\theta_I$ are optimized through minimizing the error of the predicted action $\hat{a}_t$ and the actual action $a_t$:

$$J(\theta_I) = (I(\cdot \mid s_t, s_{t+1}; \theta_I) - a_t)^2 \tag{3}$$

The motivation behind the inverse dynamic model is that the learned features should depend only on the current action of the agent and not be affected by insignificant changes in the environment. This is a theoretical assumption that is used in curiosity-driven exploration methods such as Intrinsic Curiosity Module (ICM) Pathak et al. [2017] and NGU Badia et al. [2019]. Different from the previous work, we integrate the inverse dynamic model into bisimulation metric as the measure of state discrepancy.

**Definition 2.** *(Inverse Dynamic Bisimulation Metric) Given a policy $\pi$, the inverse dynamic bisimulation metric is defined as:*

$$
\begin{aligned}
d_{inv}(s_i, s_j) = \left| r_i^\pi - r_j^\pi \right| + \gamma W_2(d_{inv})(\mathcal{P}^\pi(\cdot \mid \mathbf{s}_i), \mathcal{P}^\pi(\cdot \mid \mathbf{s}_j)) \\
+ \gamma \| I(\cdot \mid s_i, s_{i+1}) - I(\cdot \mid s_j, s_{j+1}) \|_1
\end{aligned}
\tag{4}
$$

*where $r_i^\pi = \mathbb{E}_{a \sim \pi}[R(s_i, a)]$, $\mathcal{P}^\pi(\cdot \mid s_i) = \mathbb{E}_{a \sim \pi}[\mathcal{P}(\cdot \mid s_i, a)]$ and $I(\cdot \mid s_i, s_{i+1}) = a_i$.*

In contrast to the bisimulation metric defined in Definition 1, our approach incorporates the discrepancy in action outcomes from the inverse dynamic model, thereby encouraging more effective exploration. The ablation study on inverse dynamic is also conducted in experiments. Our approach involves learning the inverse dynamic bisimulation metric through the iterative process of gradient descent. Notably, we provide a rigorous proof in Theorem 1 demonstrating the convergence of our method to a fixed point under certain assumptions. Assume that our inverse dynamic bisimulation metric is parameterized with $\phi$, to train the metric function towards Equation (4), we draw batches of state pairs, and minimize the mean square error:

$$
\begin{aligned}
J(\phi) = (\| d_{inv}(s_i, s_j; \phi) \|_1 - |r_i - r_j| \\
- \gamma W_2(\mathcal{P}(\cdot \mid \bar{s}_i, a_i; \eta), \mathcal{P}(\cdot \mid \bar{s}_j, a_j; \eta)) \\
- \gamma \| I(\cdot \mid \bar{s}_i, s_{i+1}; \theta_I) - I(\cdot \mid \bar{s}_j, s_{j+1}; \theta_I) \|_1)^2
\end{aligned}
\tag{5}
$$

where $r$ are rewards, $\bar{s}$ denotes state with stop gradients, $\mathcal{P}(\cdot \mid s, a; \eta)$ indicates probabilistic dynamics model parameterized with $\eta$ which outputs a Gaussian distribution and $I(s_t, s_{t+1}; \theta_I)$ is the inverse dynamic model parameterized with $\theta_I$ which outputs predicted action. Note that we use the 2-Wasserstein metric $W_2$[2] in Equation (5) following Zhang et al. [2020] since the $W_2$ metric has a convenient closed form: $W_2 \left( \mathcal{N}(\mu_i, \Sigma_i), \mathcal{N}(\mu_j, \Sigma_j) \right)^2 = \| \mu_i - \mu_j \|_2^2 + \left\| \Sigma_i^{1/2} - \Sigma_j^{1/2} \right\|_{\mathcal{F}}^2$, where

---

[2]The analysis of difference with 1-Wasserstein metric is in Appendix B.

$\| \cdot \|_{\mathcal{F}}$ is the Frobenius norm. For all other distances we continue using the $\ell_1$ norm, the detailed discussion on choice of norm and ablation study can be found in Appendix D.7.

**Potential-based Exploration Bonus.** Our approach defines the inverse dynamic bisimulation metric as a potential function that distills the discrepancy between states into differences of potentials, which can serve as an exploration bonus to encourage exploration.

**Definition 3.** *(Inverse Dynamic Bisimulation Metric-based Potential Function) Given an initial state $s_0$, : $\Phi : S \rightarrow \mathbb{R}$ can be written as:*

$$\Phi(s) = d_{inv}(s, s_0) \tag{6}$$

*where $d_{inv}$ is the inverse dynamic bisimulation metric in Definition 2.*

Based on Equation (1) in PBRS method Ng et al. [1999], we define our shaping reward function as:

$$\mathcal{F}(s_t, a, s_{t+1}) = \gamma d_{inv}(s_{t+1}, s_0) - d_{inv}(s_t, s_0) \tag{7}$$

As shown in Equation (7), our potential function effectively converts the state discrepancy into a reward signal to incentivize exploration. As a result, the agent is rewarded more when it encounters novel states during training, as outlined in Algorithm 1 in Appendix E.

## 5 Theoretical Analysis

LIBERTY promotes exploration by utilizing potential function (6) to distill the differences between states into discrepancies of potentials, this raises the question of how the potential-based exploration bonus simultaneously enhances training efficiency while ensuring policy invariance. In this section, we present theoretical analysis[3] that explores the connection between the potential function as value difference bound and the optimal value function, which explains the question above.

First, we show that our inverse dynamic bisimulation metric converges to a fixed point, starting from the initialized policy $\pi_0$ and converging to an optimal policy $\pi^*$.

**Theorem 1.** *Let $\mathfrak{met}$ be the space of bounded pseudo-metrics on state space $\mathcal{S}$, $\gamma \in [0, 1)$ and $\pi$ a policy that is continuously improving. Define $\mathcal{H} : \mathfrak{met} \mapsto \mathfrak{met}$ by:*

$$\begin{aligned} \mathcal{H}(d, \pi)(s_i, s_j) = |r_{s_i}^{\pi} - r_{s_j}^{\pi}| + \gamma W(d)(\mathcal{P}_{s_i}^{\pi}, \mathcal{P}_{s_j}^{\pi}) \\ + \|I(\cdot \mid s_i, s_{i+1}) - I(\cdot \mid s_j, s_{j+1})\|_1 \end{aligned} \tag{8}$$

*Then $\mathcal{H}$ has a least fixed point $\tilde{d}$ which is a inverse dynamic bisimulation metric.*

Bisimilarity is based on a recursive computation of future transition probabilities and rewards, which is closely linked to the value function. The following result demonstrates that the value difference is bounded by our inverse dynamic bisimulation metric, which also implies that the closer two states are in terms of $d_{inv}$, the more likely they are to share the same optimal actions.

**Theorem 2.** *(Value difference bound) Given any two states $s_i, s_j \in \mathcal{S}$ in an MDP $\mathcal{M}$, let $V^{\pi}(s)$ be the value function of policy $\pi$, we can get:*

$$|V^{\pi}(s_i) - V^{\pi}(s_j)| \leq d_{inv}(s_i, s_j) \tag{9}$$

*where $d_{inv}$ is a inverse dynamic bisimulation metric.*

So agents are encouraged to explore states with higher value difference (TD error), which significantly boost training efficiency. We also detail the relation between potential function $d_{inv}(s, s_0)$ and optimal value function $V^*(s)$.

**Theorem 3.** *The potential function $d_{inv}(s, s_0)$ is an approximation of the absolute value of optimal value function $V^*(s)$.*

**Theorem 4.** *Suppose that the shaping reward function $\mathcal{F}$ takes the form of Equantion (1), the optimal value function of the modified MDP $\mathcal{M}'$, the potential function $\Phi(s)$ and the optimal value function of original MDP $\mathcal{M}$ holds the condition that:*

$$V_{\mathcal{M}'}^*(s, a) = V_{\mathcal{M}}^*(s, a) - \Phi(s) \tag{10}$$

---

[3]All the detailed proof can be found in Appendix C.

**Remark.** Theorem 4 introduces the relation between optimal value function of the original MDP $\mathcal{M}$ and optimal value function of the modified MDP $\mathcal{M}'$, which explains the necessity of a good choice of potential function. Theorem 3 provides the reason why our potential function can accelerate training efficiency. Since $d_{inv}(s, s_0)$ is an approximation of absolute value of optimal value function, the value function of modified MDP $V^*_{\mathcal{M}'}$ can be learned efficiently only by focusing the non-zero values. In essence, Theorem 3 and 4 analyze how our method promotes efficiency from the view of the learning of value function. By incorporating such a potential function, we can enhance exploration and improve training efficiency, leading to faster convergence during the training process.

## 6 Experiments

The overall objective of our experiments is to evaluate the performance of LIBERTY comparing with other competitive methods, we conduct experiments on various settings of 9 continuous control tasks and 8 discrete-action games to assess the robustness and scalability of our algorithm. The implementation details can be found in Appendix E. The code is available at `https://github.com/Mingle0228/liberty`.

**Baselines.** We compare our method with several baselines and state-of-the-art methods including exploration-based methods and potential-based reward shaping methods (The detail comparison can be found in Appendix E). The exploration methods include famous benchmarks in curiosity-driven exploration, ICM Pathak et al. [2017], RND Burda et al. [2018] and NGU Badia et al. [2019] which is the extension of RND to achieve long term exploration. We also compare with RIDE Raileanu and Rocktäschel [2020] which also uses the state difference in the latent space. As for the PBRS method, we benchmark against DPBA Harutyunyan et al. [2015], a variant of PBRS method and the shaping reward is defined as the difference between potential values transformed from an arbitrary reward function.

### 6.1 Continuous Control

Firstly, we evaluate our agent in the MuJoCo continuous control[4] environment Duan et al. [2016] and use PPO Schulman et al. [2017] as the baseline RL algorithm. The six tasks evaluated in the experiment are HalfCheetah, Hopper, Walker2d, Ant, Swimmer and Humanoid.

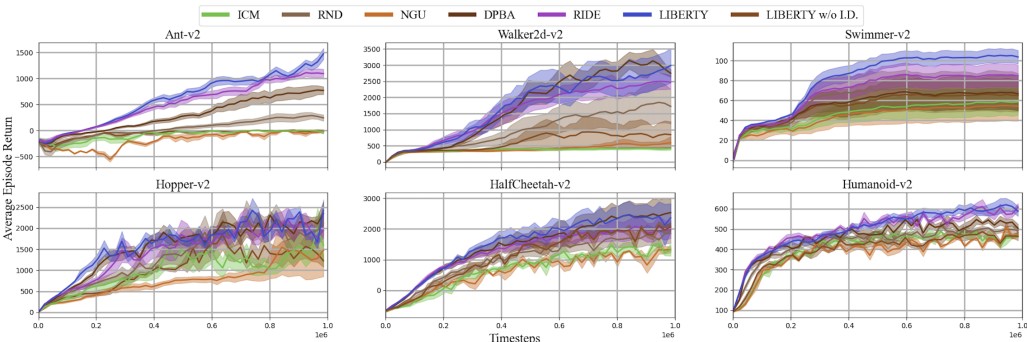

Figure 3: Comparison between LIBERTY and other approaches in the MuJoCo environments. The x-axis represents the number of steps (1e6) in training. The y-axis represents the average episode return over the last 100 training episodes (standard deviations in shade). All of the experiments were run using 10 different seeds.

The overall comparisons are presented in Figure 3, our method achieves the best rewards among all tasks, showing its superiority over continuous control tasks. RIDE obtains second best performance in five tasks which indicates the benefit of distance-based states novelty in latent space. Note that the variance of DPBA is large across all the tasks which verifies that sometimes the shaping reward may mislead the agent from optimal policy. Other exploration methods like ICM, RND and NGU

---

[4]For continuous control, we also evaluate LIBERTY in more challenging goal-conditioned tasks: `FetchPush`, `FetchPickAndPlace`, `FetchSlide`, the results can be found in Appendix D.

struggles behind because in the standard reward setting their exploration sometimes fail with rich extrinsic reward from the environment.

Table 1: Quantitative results comparison between LIBERTY and other baseline methods in different environments of Mujoco with the delayed reward setting.The best and the runner-up results are (**bold**) and (underline)

| Methods | Delay = 10 | | | | | |
|---|---|---|---|---|---|---|
| | HalfCheetah | Hopper | Walker2d | Ant | Humanoid | Swimmer |
| ICM | $1374 \pm 368$ | $1258 \pm 325$ | $1127 \pm 225$ | $-105 \pm 43$ | $462 \pm 54$ | $27 \pm 11$ |
| RND | $1694 \pm 495$ | $1976 \pm 458$ | $1405 \pm 262$ | $143 \pm 17$ | $532 \pm 29$ | $32 \pm 15$ |
| NGU | $1180 \pm 513$ | $989 \pm 262$ | $1275 \pm 480$ | $-164 \pm 35$ | $413 \pm 78$ | $24 \pm 12$ |
| RIDE | $2467 \pm 456$ | $1876 \pm 431$ | $1651 \pm 325$ | $92 \pm 31$ | $570 \pm 45$ | $65 \pm 16$ |
| DPBA | $1514 \pm 365$ | $2103 \pm 129$ | $1997 \pm 115$ | **$592 \pm 67$** | $518 \pm 23$ | $43 \pm 17$ |
| LIBERTY | **$2973 \pm 437$** | **$2479 \pm 315$** | **$2766 \pm 487$** | $292 \pm 68$ | **$681 \pm 73$** | **$73 \pm 21$** |
| LIBERTY w/o I.D. | $1783 \pm 412$ | $1676 \pm 275$ | $1732 \pm 392$ | $131 \pm 22$ | $505 \pm 37$ | $46 \pm 11$ |
| Methods | Delay = 40 | | | | | |
| | HalfCheetah | Hopper | Walker2d | Ant | Humanoid | Swimmer |
| ICM | $919 \pm 199$ | $857 \pm 175$ | $697 \pm 172$ | $-213 \pm 27$ | $403 \pm 34$ | $13 \pm 7$ |
| RND | $1276 \pm 387$ | **$1683 \pm 338$** | $968 \pm 168$ | $71 + 15$ | $483 \pm 25$ | $17 \pm 11$ |
| NGU | $1028 \pm 405$ | $879 \pm 155$ | $997 \pm 280$ | $-198 \pm 27$ | $387 \pm 27$ | $11 \pm 6$ |
| RIDE | $1798 \pm 355$ | $1235 \pm 269$ | $1025 \pm 282$ | $63 \pm 18$ | $468 \pm 23$ | **$32 \pm 11$** |
| DPBA | $883 \pm 275$ | $1382 \pm 85$ | $1016 \pm 129$ | $105 \pm 31$ | $405 \pm 15$ | $9 \pm 3$ |
| LIBERTY | **$2039 \pm 315$** | $1612 \pm 215$ | **$1921 \pm 372$** | **$142 \pm 45$** | **$566 \pm 35$** | $31 \pm 13$ |
| LIBERTY w/o I.D. | $1231 \pm 253$ | $1213 \pm 207$ | $1012 \pm 358$ | $58 \pm 13$ | $455 \pm 27$ | $17 \pm 8$ |

**Delayed Reward Setting.** We use the delayed reward setting Zheng et al. [2018] in MuJoCo environments to increase the difficulty for agent learning with sparse reward. Specifically, for the delayed reward setting, the accumulated reward is only given every 10, 20, 30 or 40 steps, so the extrinsic reward is less informative where exploration is much necessary in this setting. The results are demonstrated in Table 1 (Full table in Appendix D). In the delayed reward setting, LIBERTY achieves the best performance in 9 cases out of 12 delayed reward tasks. This indicates that with sparse reward, LIBERTY still attains efficient exploration so that the performance does not drop. Due to only receiving delayed rewards, the performance of DPBA drops with the increase of the delay period. RIDE and RND can achieve better results than DPBA, because they can provide more exploration to the agent at each step. And the performance of curiosity-driven methods are almost at the same level with delayed rewards. The result further demonstrates that the potential-based exploration of LIBERTY encourages the agent to efficiently explore in the environment even with only delayed rewards.

**Reward-free Exploration.** As for the investigation of reward-free exploration, we discretize the state-space into bins and compare the number of bins explored, in terms of coverage percentage. An agent being able to visit a certain bin corresponds to the agent being able to solve an actual task that requires reaching that certain area of the state space. Thus, it is important that a good exploration method would be able to reach as many bins as possible. We evaluate all six tasks in the MuJoCo environments and train the agent using the intrinsic reward alone. The result of HalfCheetah is presented in Figure 4, and the results of other tasks can be found in Appendix D. In the HalfCheetah environment, which has the most complex dynamics compared to the other environments, the state space is discretized into 100 bins. LIBERTY achieves the highest number of bins, covering approximately 72%. RIDE and NGU follow closely with approximately 69% and 65% coverage, respectively. RND outperforms ICM by almost two-fold, achieving 53% and 37% coverage, respectively.

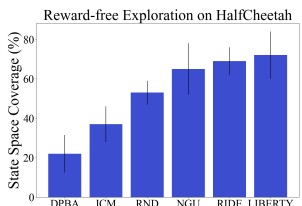

Figure 4: Results on HalfChee-tah. Error bars represent std, deviations over 10 seeds.

DPBA performs the worst, with only around 22% coverage. These results offer compelling evidence for the scalability of LIBERTY to continuous control tasks, further demonstrating its wide range of potential applications.

**Ablation Study on Inverse Dynamic.** In order to investigate the importance of inverse dynamic, we denote the variant of LIBERTY as "LIBERTY w/o I.D." which is trained without inverse dynamic. The variant uses the setting of bisimulation metric Castro [2020] only as the potential function. According to Figure 3, we can see that the performance has a significant drop without inverse dynamic, which indicates that our inverse dynamic bisimulation metric provides more effective

exploration during training, and the numerical results are presented in Table 1. The ablation results of Atari games can be found in Appendix D.

## 6.2 Atari Games

To investigate LIBERTY for high dimensional inputs and discrete actions, we also evaluate our approach on the Atari games Bellemare et al. [2013]. For the atari games, the environments chosen are designed in a way that either requires the player to explore in order to succeed, e.g. Qbert and BeamRider, or to survive as long as possible to avoid boredom, e.g. Pong and Breakout.

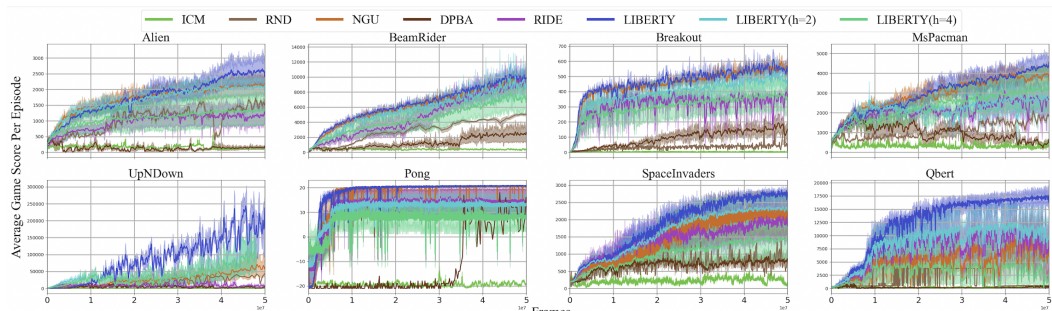

Figure 5: Comparison between LIBERTY and other approaches in the atari games. The x-axis represents the number of frames (1e7) in training. The y-axis represents the average game score per episode over the last 100 training episodes (standard deviations in shade). All of the experiments were run using 10 different seeds.

The overall performance is presented in Figure 5, LIBERTY has fastest convergence in 7 out of 8 games and achieves comparable or better results than baseline methods towards the end, which demonstrates the superiority and efficiency of our method. NGU achieves the second-best result in 6 out of 8 games, thanks to its long-term exploration. RIDE and RND yield comparable results. However, ICM and DPBA perform poorly, suggesting that the shaping reward or unnecessary exploration may sometimes mislead the agent. In Appendix D, we provide additional experiments to demonstrate the improvements of LIBERTY over various baselines. This provides further evidence of the benefits of using LIBERTY rewards in conjunction with other benchmarks.

**Ablation Study on Length of State Sequence.** The default setting is to use the distance of adjacent states as reward signal, we further investigate the performance of LIBERTY using different lengths of consecutive states to calculate the shaping reward during training. In this case, Equation (7) can be re-wrriten as $\mathcal{F}(s_t, a, s_{t+h}) = \gamma d_{inv}(s_{t+h}, s_0) - d_{inv}(s_t, s_0)$ where $h$ is the size of consecutive states for calculating the shaping reward. We demonstrate the results LIBERTY with $h = 2$ and LIBERTY with $h = 4$ in Figure 5. We can observe that as the value of $h$ increases, the performance of LIBERTY experiences a significant decline. This suggests that when the interval between states is longer, the novelty of the system decreases, resulting in less effective exploration.

## 7 Conclusion

In this work, we propose an efficient potential-based exploration framework for reward shaping, which measures the novelty of adjacent states by calculating their distance using inverse dynamic bisimulation metric. We have formulated the potential function based on our bisimulation metric and provided insightful analysis on how our shaping reward can accelerate the training speed by analyzing its relationship with the value function. Our method has proven successful in several continuous-control and discrete-action environments, providing reliable and efficient exploration performance in all the experimental domains, and showing robustness to different settings. We acknowledge that our method may encounter limitations when tackling certain tasks that require prolonged and hard exploration. Therefore, future research should concentrate on investigating the extent of these limitations and devising strategies to overcome them.

# 8 Acknowledgements

This work was supported by National Key R&D Program of China (2022YFB4501500, 2022YFB4501504), the Science and Technology Development Fund Macau SAR (0015/2019/AKP, 0031/2022/A, SKL-IOTSC-2021-2023), the Research Grant of University of Macau (MYRG2022-00252-FST), and Wuyi University Hong Kong and Macau joint Research Fund (2021WGALH14). This work was performed in part at SICC which is supported by SKL-IOTSC, University of Macau.

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
