# OpenReview forum: "Efficient Potential-based Exploration in Reinforcement Learning using Inverse Dynamic Bisimulation Metric"
_NeurIPS.cc/2023/Conference — NeurIPS 2023 poster_

### Official Review · Reviewer_AtiU · 2023-06-23

**Soundness:** 3 good
**Presentation:** 3 good
**Contribution:** 2 fair
**Rating:** 5
**Confidence:** 5

**Summary:**

This paper introduces a novel approach that combines bisimulation metrics with inverse dynamics modeling to formulate potential functions for reward shaping. The integration of these techniques offers potential-based exploration, and the paper provides theoretical analyses highlighting the benefits of this proposed method. Experiments show its robustness and scalability in various tasks.

**Strengths:**

1. A proper potential-based reward-shaping method that can preserve state differences to be based on task-specific features.
2. The main claim and the method in this paper are presented in a clear and straightforward manner, making them easy to comprehend.
3. The proposed exploration bonus does not rely on prior human knowledge.
4. Utilizing bisimulation-based metrics as an exploration bonus is interesting and worth to be investigated.

**Weaknesses:**

1. Although the empirical studies have demonstrated successes in Mujoco tasks and Atari games, it would be beneficial to conduct additional experiments that compare exploration methods. The authors should also evaluate their method on DMC [1] tasks, such as Humanoid tasks, which are known to pose more challenging exploration scenarios. Utilizing benchmarks such as URLB [6] could provide helpful insights for evaluating its exploration ability in an reward-free settings.
2. There are some confusing notations in the manuscript. For instance, in Section 3, the reward is defined as $r = \mathcal{R}(s, a)$; however, the modified reward function $\mathcal{R}' = \mathcal{R} + \mathcal{F}$ introduces a discrepancy in the arguments, where $\mathcal{F}(s, a, s') = \gamma\Phi(s') - \Phi(s)$. This inconsistency is also present in the reward function of Theorem 4 in the Appendix. To maintain clarity, the notations should be consistent throughout the entire manuscript. Additionally, Equation 29 appears unusual, as the expectation is computed by sampling the next state $s'$ while there is another $s'$ as the subscript of the max term within the expectation, which seems strange.
3. Many of the theoretical analyses (Theorem 1-3) closely resemble previous works ([2, 3, 4, 5]), which limits the novelty of these contributions. Furthermore, there appear to be issues with the proof of Theorem 4 (see in 2.), which undermines the soundness of the paper. It is important to address these concerns to strengthen the overall manuscript.

[1]: Tassa, Yuval, Doron, Yotam, Muldal, Alistair, Erez, Tom, Li, Yazhe, Casas, Diego de Las, Budden, David, Abdolmaleki, Abbas, Merel, Josh, Lefrancq, Andrew, et al. Deepmind control suite. arXiv preprint arXiv:1801.00690, 2018.

[2]: Amy Zhang, Rowan Thomas McAllister, Roberto Calandra, Yarin Gal, Sergey Levine: Learning Invariant Representations for Reinforcement Learning without Reconstruction. ICLR 2021

[3]: Norm Ferns, Prakash Panangaden, and Doina Precup. Metrics for finite markov decision processes. In UAI, volume 4, pages 162–169, 2004.

[4]: Norman Ferns, Doina Precup: Bisimulation Metrics are Optimal Value Functions. UAI 2014: 210-219

[5]: Pablo Samuel Castro: Scalable Methods for Computing State Similarity in Deterministic Markov Decision Processes. AAAI 2020: 10069-10076

[6]: Michael Laskin, Denis Yarats, Hao Liu, Kimin Lee, Albert Zhan, Kevin Lu, Catherine Cang, Lerrel Pinto, Pieter Abbeel: URLB: Unsupervised Reinforcement Learning Benchmark. NeurIPS Datasets and Benchmarks 2021

**Questions:**

1. In lines 185-189, it is unclear why exploration solely based on bisimulation metrics would lead to meaningless exploration. The bisimulation metric is derived from the reward function defined for the task. If the reward function effectively captures the task's objectives (e.g., collecting coins has a higher reward than attacking monsters), it can be assumed that the features learned by the bisimulation metric would also be meaningful. Further clarification is needed to address this potential contradiction.

2. It is unclear whether $\theta$ represents the parameters of the inverse dynamics model or the policy network. To avoid confusion, the authors should use different notations to distinguish between these two entities explicitly.

3. Equation 4 introduces the L1-norm between $a_i$ and $a_j$ as the discrepancy in action, but it is not clear why this specific norm is chosen instead of alternatives such as the L2-norm or some measurements of the distribution distance like W-distance. The authors should provide explanations or justifications for the selection of the L1-norm to enhance understanding and reasoning behind this decision.

**Limitations:**

As the proposed method requires permuting the batch to compute the metric (similar to DBC), it may require higher computational complexity (wall-clock time) compared to the other potential-based methods.

---

> ### Author Rebuttal · Authors · 2023-08-09
>
> Thank you for your insightful reviews, for the citations in the response please refer to the reference list in the global comment.
>
> *W1: The authors should also evaluate their method on DMC tasks... are known to pose more challenging exploration scenarios...*
>
> **Response**: Thank you for your suggestions, the DMC tasks and URLB benchmark are very interesting and  we are eager to evaluate our algorithm on these benchmarks if time permits.  The importance of the challenging exploration scenarios has also been discussed in [M] and [N], where the MuJoCo environment with a delayed reward setting is acknowledged as a challenging benchmark for exploration problems. To further address your concern, we additionally include results in another challenging exploration scenario like the goal-conditioned environments, suggested by the 3rd Reviewer pk3V, where the reward is only given when reaching the goal. The results can be found in the Figure 3 in global comment, we can see that our method still achieves best performance in the challenging tasks of widely used [K,L] goal-conditioned environments.
>
> *W2: There are some confusing notations in the manuscript... in Section 3, the reward is defined...This inconsistency is also present in the reward function of Theorem 4...Equation 29 appears unusual...*
>
> *Q2: It is unclear whether $\theta$ represents the parameters of the inverse dynamics model or the policy network. To avoid confusion, the authors should use different notations to distinguish between these two entities explicitly.*
>
> **Response**: Thank you for your valuable comments. We use $\theta_I$ to distinguish the parameters of the inverse dynamics model from policy network, and we will consider selecting another notation to replace $\theta_I$ to avoid confusion. We will make all the notations of the reward function in sec 3 and thm 4 consistent, e.g. $\mathcal{R}^{\prime}(s,a,s^{\prime})=\mathcal{R}(s,a,s^{\prime})+\mathcal{F}(s,a,s^{\prime})$. Furthermore, considering that Eq(29) represents the Bellman optimality equation and $V_{M}^*(s)$ denotes the optimal value function which means $\max_{s^{\prime} \in \mathcal{S}} V_{M}^*(s^{\prime}) = V_{M}^* (s^{\prime})$, we will eliminate the max term to alleviate the potential confusion, so Eq(29) equals to  $V_M^*(s, a)=\mathrm{E}_{s^{\prime}\sim \mathcal{P}(\cdot \mid s, a)}[\mathcal{R}(s, a, s^{\prime})+\gamma  V_M^*(s^{\prime})]$ The notation problem will be fixed in the revised version.
>
> *W3: Many of the theoretical analyses (Theorem 1-3) closely resemble previous works ([2, 3, 4, 5]), which limits the novelty of these contributions...*
>
> **Response**: Previous work [2, 3, 4, 5] (citations refer to the review) mainly emphasize state representation learning and the computation and evaluation of bisimulation metrics. Specifically, [2] uses the bisimulation metric to learn the state representation, [5] just proposes the algorithm for computing on-policy bisimulation metrics. [3,4] proposes the theory framework for bisimulation metric and the the theory guarantee of [2] and [5] are also based on [3,4]. However, these approaches differ significantly from our work, which focuses on boosting exploration under the reward shaping framework. We are the first to propose the **inverse dynamic** bisimulation metric for exploration and the novelty is evidenced by other 3 reviewers pk3V ,dnFb and naKC (strength points). All the theoretical analysis are based on **reward shaping** framework, which significantly different from **representation learning** and **metric computation**.  Specifically, our metric incorporates the inverse dynamic based on the bisimulation metric, making thm 1 crucial in ensuring the convergence of our approach. Theoretical analysis in Thm. 2-4 mainly focus on how our method effectively promotes **efficient** exploration .  We will improve the emphasis on our contribution in Sec. 5, and the notation issue in Thm. 4 is fixed in the revised version.
>
> *Q1: ..it is unclear why exploration solely based on bisimulation metric would lead to meaningless exploration...features learned by the bisimulation metric would also be meaningful..*
>
> **Response**: Firstly, we mean the **shaping reward** calculated solely based on the bisimulation metric may  result in meaningless exploration bonus. While we acknowledge the value of the features learned through the bisimulation metric within the framework of **state representation learning**, it's important to clarify that we only use the metric for calculating state differences as a shaping reward to assist exploration, rather than using it for training the state representation. We mean that exploration bonus calculated by the bisimulation metric can't detect state changes resulting from other environmental factors. For example, the agent Mario is navigating a level  with random moving monsters and changing background, it will visit a vast number of different states and collect lots of cumulative reward without taking actions. The agent will learn to stay in the same position without taking any useful actions since the shaping reward is still high in this case. So we introduce the inverse dynamic module in the metric to identify whether the state changes are caused by actions, so the agent can explore more effectively.
>
> *Q3: Equation 4 introduces the L1-norm between $a_i$ and $a_j$ as the discrepancy in action, but it is not clear why this specific norm is chosen instead of alternatives such as the L2-norm or some measurements of the distribution distance like W-distance...*
>
> **Response**: We followed  previous famous work([E,F]) which includes the inverse dynamic module.  L1-norm is the lowest computational cost since the output of the action in inverse dynamic module is a deterministic scalar or vector in RL environments. We believe the more detailed discussion about the choice of L1-norm or L2-norm is a direction for future work in the specific algorithm and environment.

---

> > ### Comment · Reviewer_AtiU · 2023-08-15
> >
> > After carefully reviewing all the responses from the reviewer, I have several questions and suggestions for improvement:
> >
> > 1.
> >
> > > $V_M^*\left(s\right)$ denotes the optimal value function which means $\max _{s^{\prime} \in S} V_M^*\left(s^{\prime}\right)=V_M^*\left(s^{\prime}\right)$
> >
> > The statement regarding $\max _{s^{\prime} \in S} V_M^*\left(s^{\prime}\right)$ and $V_M^*\left(s\right)$ is incorrect. $V_M^*\left(s\right)$ represents the optimal value of state s, while $\max _{s^{\prime} \in S} V_M^*\left(s^{\prime}\right)$ represents the maximum value over all states. Thus, they are not equivalent.
> >
> > 2.
> >
> > > All the theoretical analyses are based on reward shaping framework, which is significantly different from representation learning and metric computation.
> >
> > Theorems 1-3 (specifically from Line 498 to Line 550 in Appendix C) are not relevant to reward shaping. This should be clarified.
> >
> > 3.
> >
> > > For example, the agent Mario is navigating a level with random moving monsters and changing background, it will visit a vast number of different states and collect lots of cumulative reward without taking actions.
> >
> > How the agent can collect cumulative rewards without taking any actions, even if the background keeps changing? Rewards are typically obtained by collecting coins, hitting monsters, or reaching the flag. It is unclear how rewards can be collected by staying in the same position. Please provide further clarification on this point.
> >
> > 4.
> >
> > > We followed previous famous work([E,F]) which includes the inverse dynamic module.
> >
> > I was unable to find the specific module referred to in paper [F] (specifically used L1-norm settings). I would suggest the authors provide the exact Equation or Lines of descriptions for more information.
> >
> > Besides, as far as I know, [1] has already used bisimulation as a bonus term. From this perspective, the main contribution of this paper is to integrate the inverse dynamics model into bisimulation to design exploration bonuses. Consequently, it is important to clearly emphasize the specific reasons for choosing each module, particularly the non-trivial design of the inverse dynamic module.
> >
> > [1]: Dadashi R, Rezaeifar S, Vieillard N, et al. Offline reinforcement learning with pseudometric learning[C]//International Conference on Machine Learning. PMLR, 2021: 2307-2318.
> >
> > 5. I partially agree with the Q3 from Reviewer naKC.
> >
> > > **Response to Q3 from naKC**: The shaping reward will have no impact on training, so the agent stops exploring after the convergence of optimal policy
> >
> > No, it just means the bisimulation metric converges to its fixed point, instead of meaning that the potential function is zero. $\gamma d_\text{inv}(s',s_0) - d_\text{inv}(s,s_0)$ would still be large if $s$ and $s'$ are totally different states. Then this bonus will keep being non-zero even if the policy is converged.
> >
> > 6.
> >
> > > **Response to Q2 from dnFb**: So the primary focus of the shaping reward is on capturing the difference between $s_t$ and $s_{t+1}$.
> >
> > I noticed that in Equation 24, the derivation shows that $|V^{\pi}(s)-V^{\pi}(s_0)|=|V^{\pi}(s)|-C_1$. Let's say two different cases: $V^{\pi}(s)=5$ and $V^{\pi}(s)=1$, and we set $V^{\pi}(s_0)=2$. The values should be $|5-2|=3$ and $|1-2|=1$ respectively. However, the equation suggests $|V^{\pi}(s)|-C_1$. I am confused about this discrepancy. I do hope the authors can help me to address this.
> >
> > Overall, I believe addressing these questions and suggestions will help improve the clarity and accuracy of the paper. I will keep the score as-is.

---

> > > ### Author Response · Authors · 2023-08-15
> > > **Emphasizing the contribution and addressing notation issues**
> > >
> > > Thanks for your comment, we believe the main question of you is the contribution of our work. Apart from including the inverse dynamic module in bisimulation metric, we want emphasize that our key contribution is that we are the **first** exploration method to address
> > >
> > > **(1)** no guarantee convergence of optimal policy,
> > >
> > > **(2)** lack scalability,
> > >
> > > **(3)** reliance on prior knowledge,
> > >
> > > and promoting learning efficiency (achieves SOTA in competitive envs). Please refer to line 28-46, table 3 in Appendix E and Appendix D.3 and D.4 where we had clarified and verified the weakness of typical exploration methods. The contribution is also acknowledged by reviewer dnFb and pk3V. the detailed response is as follows.
> > >
> > > Q1: ..$\max _{s^{\prime} \in S} V_M^*(s^{\prime})$ and $V_M^*(s)$  are not equivalent..
> > >
> > > A1: Yes, $\max _{s^{\prime} \in S} V_M^*(s^{\prime})$ is not equivalent to $V_M^*(s)$,  we means that  Eq(29) is based on **bellman optimality function** where the max term over states should be omitted: $V_M^*(s)=\max _a \underset{s^{\prime} \sim P}{\mathrm{E}}[R(s, a)+\gamma V_M^*(s^{\prime})]$ , the max term is chosen over actions and we will fix it in revision.
> > >
> > > Q2: Theorems 1-3 (specifically from Line 498 to Line 550 in Appendix C) are not relevant to reward shaping. This should be clarified.
> > >
> > > A2: Our shaping reward is calculated by bisimulation metric-based potential function, so the convergence of the metric (thm 1) is to ensure that our shaping reward meets the form of potential-based reward shaping(Eq (1)). Thm 2 demonstrates the property of our shaping reward of why it can encourage agents to explore states with high value difference. Thm 3 serves thm 4 by explaining how our shaping reward function changes the optimal function between the original and modified MDP. These analyses are all conducted within the framework of **potential-based reward shaping** and aim to elucidate reasons behind the superior efficiency of our shaping reward (please also refer to 1st response in reviewer pk3V).
> > >
> > > Q3: ..It is unclear how rewards can be collected by staying in the same position. Please provide further clarification on this point.
> > >
> > > A3: The reward you mentioned is the original reward function $R$ feedback from the environment. In this work, we reshape the reward calculation by $R^{\prime} = R + F$(see line 115), where $F(s_t, a, s_{t+1})=\gamma d_{i n v}(s_{t+1}, s_0)-d_{i n v}(s_t, s_0)$, when background state changes, the shaping reward $F$ collected by agent is high, thus collecting numerous rewards.
> > >
> > > Q4: ..I was unable to find the specific module referred to in paper [F]..[1] has already used bisimulation as a bonus term..it is important to clearly emphasize the specific reasons for choosing each module.
> > >
> > > A4: Please refer to the Embedding network (inverse dynamic module) in section 2 of [F], which is built upon a Siamese network [Koch et al.] using L1 norm to maiximize likelihood, besides please check the L1-norm of inverse dynamic model used in Eq (2) in [Hong et al.], and many other work using L1 norm in the module (please check section 2 and Eq(6) in [Meier et al.]. The intuitive reason behind L1 norm is the lowest computational cost since the action output is mostly **scalar** in RL envs. To further address your concern, we will add more citation on the choice of L1 and discuss its lowest computational cost in revision.
> > >
> > > Bonus term in [1] is not a "reward", which is very different from ours. We use the metric to calculate exploration reward and [1] uses it to restrict exploration without modifying the reward (the author of [1] details the difference in sec 3.1).
> > >
> > > We want to emphasize again that our **primary** contribution lies in being the first exploration method to ensure the convergence of an optimal policy and enhance learning efficiency, without relying on prior knowledge.
> > >
> > > Q5: ..this bonus will keep being non-zero even if the policy is converged...
> > >
> > > A5: The policy $\pi^*(s)$ is converged, in this case although the bonus (shaping reward $F$) remains non-zero, it won't affect the how agent choose actions since the input of the policy $\pi^*(s)$  is solely the state and not the reward, so our bonus will have no impact on training.
> > >
> > > Q6: ..I am confused about this discrepancy..
> > >
> > > A6: Sorry it's a confusion caused by the line break, we mean $d_{inv}(s, s_0) \geq |V^\pi(s)-V^\pi(s_0)| \geq |V^\pi(s)| - |V^\pi(s_0)|$, since $s_0$ is the initial state $env.reset()$ fixed in each episode, we have  $|V^\pi(s_0)| \geq constant = C_1$, thanks for pointing out the issue and we will fix in the revision.
> > >
> > > We do hope that your concerns are well addressed.
> > >
> > > * Reference
> > > > [Koch et al.] Koch G, Zemel R . Siamese neural networks for one-shot image recognition ICML 2015
> > > >
> > > > [Hong et al.] Hong Z W, Fu T J, et al. Adversarial active exploration for inverse dynamics model learning CoRL 2020
> > > >
> > > >[Meier et al.] Meier F, Kappler D, et al. Towards robust online inverse dynamics learning. IROS IEEE 2016

---

> > > > ### Comment · Reviewer_AtiU · 2023-08-17
> > > > **Response to authors**
> > > >
> > > > Thanks for the reply.
> > > >
> > > > 2. Theorem 1-3 are still quite similar to many previous works, and even if we do not consider the specific utilization of the bisimulation metric as a potential function, they still hold.
> > > >
> > > > 3. My concern is that if we use the previous bisimulation metric (such as DBC or MICo) as a potential function, no matter how background changes, the shaped reward should not be affected, as the fact that bisimulation can ignore task-irrelevant information.
> > > >
> > > > 4. I checked details in [F] carefully, "To avoid such meaningless exploration, given two consecutive observations, we train a Siamese network (Bromley et al., 1994; Koch et al., 2015) f to predict the action taken by the agent to go from one observation to the next (Pathak et al., 2017)." Notably, the architecture is a Siamese network, while the training objective follows [Pathak et al., 2017].  Further, Eq (2) in [Hong et al.] is L_I loss instead of L_1, L_I here means the information, not L1-norm. Please check Eq(3) carefully, they still use L2-norm instead of L1-norm.
> > > >
> > > > [Pathak et al., 2017]: Pathak D, Agrawal P, Efros A A, et al. Curiosity-driven exploration by self-supervised prediction[C]//International conference on machine learning. PMLR, 2017: 2778-2787.
> > > >
> > > > 5. A further question is: the fixed point of bisimulation is relevant to its underlying policy, while bisim-metric changing will cause the reward function to change accordingly. And furthermore, a changed reward will result in a different optimal policy. Then how can we assure the policy is converged? This is a very different setting than bisim as representation, where we can learn policy and bisim iteratively. (Please check Appendix A in DBC's paper).
> > > >
> > > > Given the above questions are unsolved, I would suggest a rejection. If all questions can be addressed carefully, I would reconsider the score.

---

> > > > > ### Author Response · Authors · 2023-08-17
> > > > >
> > > > > Thank you for your comment, we would like to provide a more detailed response for your questions as follows:
> > > > >
> > > > > Q2:Thm 1-3 similar to many previous works,  even if ..
> > > > >
> > > > > A2: We add **inverse dynamic module** upon the bisim-metric compared with previous work.  With the inclusion of an additional term, it is important and different to re-examine the analysis of all properties based on the modified bisim-metric. We had also summarized the issues of  typical exploration methods (line 34-46), and addressing these issues is our main contribution. The similarity of thms 1-3 to previous work on bisim-metric doesn't diminish our contribution. The inclusion of an **inverse dynamic module** sets our approach apart and adds a distinct perspective to the field. Thms 1-3 just serve as explanation why our method is more efficient, which is verified by extensive experiments.
> > > > >
> > > > > Q3:..if we use previous bisim-metric (DBC or MICo) as a potential..the shaped reward should not be affected..
> > > > >
> > > > > A3:When background changes, agent will receive shaped reward calculated by bisim-metric without taking actions, it's called "meaningless exploration" also pointed out in [F]. As for your concern, previous bisim-metric (DBC or MICo) still provides shaped reward in this scenario, a key point is that the transition function within the metric is estimated by neural network.**Only when the trained metric is converged**, the shaped reward may remain unaffected (near zero) when background changes. In other words, during the initial training phase, especially in sparse reward envs, shaped reward will have a large impact on the choice of actions,  the state difference caused by background-changes evaluated by bisim metric-only is significant before the metric is converged. Therefore, the integration of the inverse dynamic module is crucial by encouraging agents to take diverse actions, which is already evidenced in our ablation study (line 324-329).
> > > > >
> > > > > Q4:..Please check Eq(3) carefully, they use L2-norm instead of L1-norm..
> > > > >
> > > > > A4: We are sorry about the mistake of missing Eq (3) in [Hong et al.]. In order to address your concern,  we had carried the ablation experiment on L1 norm and L2 norm with 5 different seeds till now, the results is almost the same in both discrete-action envs and continuous-action envs, and we will include the results with more seeds (10 or 20) and add a discussion on the choice in the revision:  Actions are scalar or low dimensional vectors in most RL environments. In discrete-action env, such as Breakout, the action space is scalar so *diff(L1,L2)*=0. For the case of continuous-action, such as the 4-dimensional vector action space in the *FetchReach* env where the actions' value range is[-1,1], since the action's value range is small ,the difference *diff(L1,L2)* is almost negligible, so we choose low computational cost L1 norm. We believe when it comes to complex env in real life, the choice of specific norm is good extension for future work.
> > > > >
> > > > > Q5:..changed reward will result in a different optimal policy.. the policy is converged? ..different setting bisim as representation..
> > > > >
> > > > > A5:
> > > > > > .. a changed reward will result in a different optimal policy..
> > > > >
> > > > > The original MDP $M$ will be modified as $M^{\prime}$ after reward shaping, so there is an issue that changed reward will have an influence on the optimal policy of MDP $M$. However, the famous work PBRS had **proven** that if shaping reward is potential-based, the optimal policy of $M$ and $M^{\prime}$ stays the same (see line 113-126), so our method assures the invariance of optimal policy which means the shaping reward won't affect the optimal policy of $M$. Since it takes a long time for policy to converge in sparse reward envs, our shaping reward just serves as accelerating the learning speed of optimal policy by promoting efficient exploration.
> > > > >
> > > > > > ..different setting than bisim as representation..we can learn policy and bisim iteratively(check DBC)
> > > > >
> > > > > In the case of DBC, the policy is trained initially, followed by the training of the bisim-metric encoder. The newly encoded states are then incorporated into the policy training (see Algorithm 1 in DBC). Similarly, we first train the policy and subsequently train the inverse dynamic bisimulation metric (shaping reward function). The newly generated shaped rewards are then utilized in the policy training. So DBC and our work follows a similar iterative process.
> > > > >
> > > > > DBC use the policy improvement theorem assumption (see Eq (11) in Appendix A of DBC) to guarantee that their policy will converge and it's an assumption we also used in theorem 1, if a policy $\pi$ is continuously improving, the convergence of the optimal policy is guaranteed.
> > > > >
> > > > > In a nutshell, our shaping reward won't affect the optimal policy of the original MDP and the optimal policy convergence is under the same assumption of DBC. Note that all other typical exploration methods (ICM, RND,NGU,RIDE) lack the guarantee of policy invariance and convergence of optimal policy.

---

> > > > > > ### Comment · Reviewer_AtiU · 2023-08-18
> > > > > > **Thanks for the explanations**
> > > > > >
> > > > > > Thank you for the response. I believe that most of the questions have been answered. Although I still think that the theoretical innovation of this paper is not particularly outstanding, I do not object to accepting the paper if the author can make modifications according to the reviewers' suggestions. I think the score for this paper should be between 4-5; however, I believe that bisimulation-related methods have great potential, so I have decided to adjust the score to 5.

---

> > > > > > > ### Author Response · Authors · 2023-08-18
> > > > > > >
> > > > > > > Thanks again for the valuable feedback. We will address all the notation issues during the review phase and incorporate a discussion on the choice of L1 norm as well as an ablation experiment comparing L1 and L2 norms in the revised version. We are confident that by addressing these issues, the quality of our paper will be significantly enhanced.

---

### Official Review · Reviewer_pk3V · 2023-07-04

**Soundness:** 3 good
**Presentation:** 3 good
**Contribution:** 3 good
**Rating:** 7
**Confidence:** 4

**Summary:**

This paper focuses on the topic of reward shaping in reinforcement learning to encourage exploration. Different from previous methods that heavily rely on the count-based episodic term in the exploration bonus, they provide an end-to-end potential-based exploration bonus. This paper proposes to use the bisimulation metric in potential-based reward shaping. Specifically, they propose the Inverse Dynamic Bisimulation Metric to avoid meaningless exploration that is not caused by actions. They provide rigorous proof demonstrating the convergence of their method to a ﬁxed point under certain assumptions. Experimental results on mujoco locomotion tasks and Atari games show that their method outperforms other reward-shaping algorithms by a large margin.

**Strengths:**

1.	The paper is well-written and easy to follow. Figure 1 gives an illustrative example to explain the meaning of the exploration bonus.
2.	The idea of using bisimulation metric to identify the difference between states is a novel idea, which is intuitive but has not been explored before.
3.	Experiments on both continuous control and discrete Atari game are comprehensive and thorough.


**Weaknesses:**

1.	There are 4 theorems in the main context without enough discussion about their necessity and importance, particularly theorem 3 and theorem 4.
2.	The algorithm may need to be evaluated on sparse reward settings, for example, goal-conditioned tasks. The delayed reward setting of mujoco is still not challenging enough.
3.	A minor thing: Curves in Figure 5 could be smoothed for better visualization.


**Questions:**

1.	The observation in the Atari experiments shows that increasing the horizon h from 1 to 2 causes a signiﬁcant decline of the proposed method. Does this mean that the algorithm is unstable? In some real-world tasks or complicated tasks, the influence of action may be delayed and the consecutive states may have no differences. Therefore, we may need to select a proper h to calculate the exploration bonus. If h has such a large influence, sometimes using a fixed h may also cause problems.
2.	Reward shaping that encourages exploration is usually used for goal-conditioned tasks that have very sparse rewards. I am curious about the performance of the proposed method in such scenarios. The mujoco locomotion and Atari game environment may not be challenging enough.


**Limitations:**

One potential limitation may come from Potential-based reward shaping, which is the basis of the proposed method. Sometimes it is hard to select the horizon to calculate the difference between states. The experimental results also show that the algorithm is sensitive to the horizon.

---

> ### Author Rebuttal · Authors · 2023-08-09
>
> Thank you for your insightful reviews, we had included the required experiments in the pdf of global comment regarding to W2 and Q2. For all the references mentioned in the response, please find the reference list in the global comment.
>
> *W1: There are 4 theorems in the main context without enough discussion about their necessity and importance, particularly theorem 3 and theorem 4*
>
> **Response**: The theoretical analysis is critical in supporting the evidence of our contribution (see line 60-66). It serves to explain how our shaping reward guarantees policy invariance and enables more efficient exploration when compared to other exploration methods. The purpose of Theorem 1 is to demonstrate that our inverse dynamic bisimulation metric converges to a fixed point, ensuring that it does not interfere with the convergence of the policy. Theorem 2 provides compelling evidence supporting the intuition behind the bisimulation metric and explains Figure 2, which serves as a guarantee that our metric establishes a bound on value differences. Theorem 4 introduces the relation between optimal value function of the original MDP $M$ and optimal value function of the modified MDP $M^{\prime}$ : $V^*_{M^{\prime}}=V_{M}^*-\Phi(s)$, where $\Phi(s)$ is the potential function, which explains the **necessity** of a good choice of potential function. And Theorem 3 provides the reason why our potential function $d_{inv}$ can accelerate training efficiency. Since $d_{inv}$ is an approximation of absolute value of optimal value function, the value function of modified MDP $V^*_{M^{\prime}}$ can be learned efficiently only by focusing the non-zero V-values. In a nutshell, Theorem 3 and 4 analyze how our method promotes efficiency from the view of the learning of value function. We are the first work in achieving exceptional exploration capabilities while also boosting training efficiency by incorporating value function learning. The analysis of four theorems is imperative to substantiate the use of the term **efficient** in the title. Thank you for the comment, we will further enhance the discussion in the revised version.
>
> *W2: The algorithm may need to be evaluated on sparse reward settings, for example, goal-conditioned tasks. The delayed reward setting of mujoco is still not challenging enough*
>
> *Q2: Reward shaping that encourages exploration is usually used for goal-conditioned tasks that have very sparse rewards. I am curious about the performance of the proposed method in such scenarios. The mujoco locomotion and Atari game environment may not be challenging enough*
>
> **Response**: Thanks you for your suggestion, the atari games are the one of the most acknowledged baselines in exploration methods (ICM[E], RND[J], NGU[F]), and the mujoco benchmark is one of the most widely used [M] for continuous control problem (the actions are continuous). The delayed reward setting makes the reward more sparse with the increasing delayed steps. As for your suggestion, we include the results of widely used [K,L] goal-conditioned environments (the robot-arm control envs) FetchPickAndPlace, FetchSlide and FetchPush. The results can be found in Figure 3 in the pdf of global comment, we can see that our method LIBERTY still achieves the best performance in the challenging tasks of goal-conditioned environments compared with other baselines, and we will include the results in revision.
>
> *W3: A minor thing: Curves in Figure 5 could be smoothed for better visualization.*
>
> **Response**: Thank you for the suggestion, we will fix the issue in the revised version.
>
> *Q1: ...increasing the horizon h from 1 to 2 causes a signiﬁcant decline of the proposed method. Does this mean that the algorithm is unstable? In some real-world tasks or complicated tasks, the influence of action may be delayed and the consecutive states may have no differences. Therefore, we may need to select a proper h to calculate the exploration bonus. If h has such a large influence, sometimes using a fixed h may also cause problems.*
>
> *Limitations: Sometimes it is hard to select the horizon to calculate the difference between states. The experimental results also show that the algorithm is sensitive to the horizon*
>
> **Response**: Thank you for your comments, we emphasize that the study on the length of state sequence $h$ is an ablation experiment, and in our algorithm $h$ is set to 1 by default to meet the form of potential-based reward shaping (see Eq(1)). Recent studies [G,H] have explored the incorporation of diversity between neural networks and multi-agents to promote exploration. They conducted ablation studies to investigate factors that influence diversity within the system. In our method, the length of state sequence $h$ is the factor which influence the diversity between states sequence, so we carry the ablation study following [G,H].  In some real-world tasks or complicated tasks when the influence of action may be delayed and the consecutive states may have no differences, we agree with you that a fixed $h$ may not be the best fit, we can train a dynamic $h$ as a parameter using meta-gradient like the dynamic $\gamma$ in the work[I], and we think this is another interesting direction worth exploring in the future.

---

> > ### Comment · Reviewer_pk3V · 2023-08-14
> > **Response to authors**
> >
> > Thanks for addressing my concerns. I don't have further questions.

---

> > > ### Author Response · Authors · 2023-08-17
> > >
> > > Thank you for acknowledging our contribution. We believe that your valuable feedback will improve the quality of our paper.

---

### Official Review · Reviewer_dnFb · 2023-07-05

**Soundness:** 2 fair
**Presentation:** 3 good
**Contribution:** 3 good
**Rating:** 6
**Confidence:** 4

**Summary:**

This paper proposes the automatic construction of a potential function for policy-invariant reward transformation. The basic idea is adding the discrepancy in action outcomes from the inverse dynamic model to the on-policy bisimulation metric proposed by Castro [2020]. Then, the authors propose a method to train the metric named inverse dynamic bisimulation metric. The authors show that it is used as a potential function for reward shaping and prove that the proposed metric bounds the value difference. Experimental results show that the proposed method outperforms several baseline methods, such as ICM, RND, NGU, RIDE, and DPBA.

**Strengths:**

- Originality: The inverse dynamic bisimulation metric is novel, although it is a simple extension of the on-policy bisimulation metric.
- Quality: The experimental results support the claims and the proposed method.
- Clarity: The paper is written well and easy to follow.
- Significance: As the authors pointed out that the previous studies designed the potential function based on some domain knowledge. On the contrary, this study provides a new research direction for automatic construction.


**Weaknesses:**

- The proposed method is a kind of model-based reinforcement learning (RL) because it explicitly estimates the transition and reward functions. Therefore, comparing the proposed method and model-based approaches is important, but the authors do not discuss this point.

**Questions:**

Major comments:
- If my understanding is correct, the proposed method estimates the transition function explicitly. In addition, I think that the reward function is also estimated explicitly because $r_i^\pi$ is computed by $\mathbb{E}_{a \sim \pi}[\mathcal{R}(s_i, a)]$. It suggests that the proposed method can be interpreted as a model-based approach. Is the proposed method more efficient than model-based RL? Please discuss this point.
- Definition 3 is interesting, but it is unclear how $s_0$ is determined. If the simulation always starts from the same state, the potential function (6) is fine. However, it is unclear how $\Phi(s)$ works when $s_0$ is sampled from some initial state distribution.
- I do not fully understand why $d_{inv}(s, s_0)$ is a good approximation of the optimal value function because the optimal value function is not necessarily non-negative. For example, it is negative if the reward function is negative. However, $d_{inv}$ is non-negative.
- I understand that the inverse dynamic model $I: \mathcal{S} \times \mathcal{S} \to \mathcal{A}$ is widely used in this field, but it implicitly assumes that the action that makes a transition from $s_t$ to $s_{t+1}$ is determined uniquely. The inverse dynamic model does not work well if multiple actions make the same state transitions. Would you discuss what happens if the actions are redundant?
- The proposed inverse dynamic bisimulation metric (4) uses the 2-Wasserstein metric, while the on-policy bisimulation metric (2) uses the 1-Wasserstein metric. Although the difference is discussed in Appendix B, it is unclear how the difference between 1- and 2-Wasserstein metrics affects the learning process.

Minor comments:
- The authors explain the behavior of the exploration bonus reward in SuperMarioBros in Figure 1. However, the proposed method is evaluated on the MuJoCo and the Atari environments. It would be nice to show the results in SuperMarioBros.
- Line 124: I think that $\Phi: \mathcal{S} \to \mathcal{R}$ should be $\Phi: \mathcal{S} \to \mathbb{R}$.
- Line 224: $R$ -> $\mathbb{R}$.


**Limitations:**

Minor comment:
- The authors provide researchers with a broader impact of this study but do not discuss this work's potential negative social impacts.

---

> ### Author Rebuttal · Authors · 2023-08-09
>
> Thank you for your insightful reviews. For all the references mentioned in the response, please find the reference list in the global comment.
>
> *W1: ...because it explicitly estimates the transition and reward functions. Therefore, comparing the proposed method and model-based approaches is important...*
>
> *Q1: It suggests that the proposed method can be interpreted as a model-based approach. Is the proposed method more efficient than model-based RL? Please discuss this point...*
>
> **Response**: First, we clarify that our method only estimates the transition function but not the reward functions which is different with model-based approaches. Model-based approaches [D] involve learning the environment transition function $\mathcal{P}(s,a)$ and reward function $\mathcal{R}(s,a)$ to facilitate planning in policy learning. In our case, the computation of $r_i^{\pi}=\mathbb{E}_{a \sim \pi}[\mathcal{R}(s_i, a)]$ is done explicitly without the need to learn the reward function $\mathcal{R}(s, a)$ directly. This is because $\mathcal{R}(s, a)$ is derived from the environment's feedback. Consequently, we do not draw a direct comparison with model-based approaches.
>
> Our approach focuses on enhancing exploration by estimating transition functions as a component of the state difference. What sets our method apart are several key advantages. Firstly, our approach is **end-to-end**, eliminating the prerequisite of learning the environment model in advance. Secondly, our method has been mathematically proven to converge (refer to Theorem 1), while model-based approaches [D] often lack guaranteed convergence of their learned models. Thirdly, in the context of sparse reward scenarios, model-based approaches often yield reward functions that remain close to 0 for extended periods, which hampers their effectiveness in addressing sparse reward challenges. In contrast, our method excels in sparse reward problems, as evidenced in Table 1 of the experiments.
>
> *Q2: ...but it is unclear how $s_0$ is determined. If the simulation always starts from the same state, the potential function (6) is fine. However, it is unclear how $\Phi(s)$ works when $s_0$ is sampled from some initial state distribution.*
>
> **Response**: Good catch, $s_0$ is determined as *env.reset()* in the training period. For environment like autonomous driving where $s_0$ is sampled from some initial state distribution. At the begin of the episode, $s_0$ is sampled from the initial distribution, $\Phi(s)=d_{inv}(s, s_0)$ will serves the potential function to calculate the shaping reward for the transitions in this episode, when it reaches to the end of the episode, the transitions of this episode will be added into the buffer, the initial state $s_0^{new}$ will be resampled for the begin of the new episode, and the potential function is set as $\Phi(s)=d_{inv}(s, s_0^{new})$ in the new episode. It is noteworthy that different initial state $s_0$ has minimal impact on the training process. This is due to the shaping reward function, where $F = \gamma d_{inv}(s_{t+1}, s_0) - d_{inv}(s_t, s_0)$, in which the initial state $s_0$ acts as a baseline. So the primary focus of the shaping reward is on capturing the difference between $s_t$ and $s_{t+1}$.
>
> *Q3: I do not fully understand why $d_{i n v}\left(s, s_0\right)$ is a good approximation of the optimal value function because the optimal value function is not necessarily non-negative...*
>
> **Response**: Thank you for pointing out this issue, as shown the proof of Theorem 3 in Appendix C, $d_{inv}$ is an approximation of the *absolute value* of optimal value function, we will fix this typo in the revised version.
>
> *Q4: I understand that the inverse dynamic model $I: \mathcal{S} \times \mathcal{S} \rightarrow \mathcal{A}$ is widely used in this field, but it implicitly assumes that the action that makes a transition from $s_t$ to $s_{t+1}$ is determined uniquely. The inverse dynamic model does not work well if multiple actions make the same state transitions. Would you discuss what happens if the actions are redundant?*
>
> **Response**: Good question, we clarify that the uniqueness of action output in inverse dynamic module is widely acknowledged in previous work [E,F]. when multiple actions make the same state transitions, the inverse dynamic model can first output the probability of each redundant action $a_i$ as $p(a_i|s,s^{\prime})$, then a deterministic action $a_j$ is sampled as output, when the actions are continuous, the final output of action can be sampled from distributions (e.g. Gaussian). Consequently, the redundancy of actions has no impact on the training process in our method due to the uniqueness of action outputs.
>
> *Q5: ...it is unclear how the difference between 1- and 2-Wasserstein metrics affects the learning process*
>
> **Response**: As shown in Lemma 1 in Appendix B, for any two distributions $\mu, \lambda$, $W_1(\mu, \lambda) \leq W_2(\mu, \lambda)$. During the learning process, the $W_2$ metric offers a greater amount of shaping reward for the same transition. This proves particularly advantageous for exploration, especially in sparse reward settings where the external reward remains zero for the majority of the time. Furthermore, the closed-form solution for the $W_2$ metric when applied to Gaussians substantially reduces the computational cost associated with estimating the term "$W_2(d_{inv})(\mathcal{P}^\pi(\cdot \mid s), \mathcal{P}^\pi(\cdot \mid s^{\prime}))$" in comparison to $W_1$. Thanks for your comment, we will make a more detailed discussion in the revised version.
>
> *Minor comments*
>
> **Response**: Thank you for your valuable feedback. We would like to emphasize that this work does not have any potential negative social impacts. Furthermore, in the revised version, we will incorporate the results obtained from SuperMarioBros and address the issue regarding the character in Line 124 and 224.

---

### Official Review · Reviewer_naKC · 2023-07-06

**Soundness:** 2 fair
**Presentation:** 3 good
**Contribution:** 2 fair
**Rating:** 6
**Confidence:** 3

**Summary:**

This paper proposes to use inverse dynamic bisimulation metric for potential-based reward-shaping (PBRS). Specifically, the authors introduce the inverse dynamic bisimulation metric, which augments the bisimulation metric with an inverse dynamics term to account for state differences caused by actions. They then use the inverse dynamic bisimulation distance between the initial state and the current state as the potential function for PBRS. Compared to the L2 distance used in standard PBRS, inverse dynamic bisimulation metric prioritizes visitation of states with higher TD error. Moreover, this bisimulation metric enjoys numerous theoretical guarantees, including convergence to fixed-point and connections to the value difference. The authors validate the superiority of their methods compared to prior curiosity-based and potential-based exploration methods across a suite of MuJoCo and Atari tasks.

**Strengths:**

- The idea of using the bisimulation metric for potential-based exploration is original. In particular, by augmenting the bisimulation matrix with an inverse dynamic term, the agent is less incentivized to visit states with similar action outcomes.
- The experiments are substantiative and exhaustive, demonstrating an overall improvement from prior exploration methods.
- The theoretical results are solid.

**Weaknesses:**

- While using the bisimulation metric for exploration is appealing, the intuition for doing so remains largely unclear. From my understanding, the bisimulation metric offers a means to partition the state space via the similarity of reward and transition. However, it is unclear how incentivizing the agent to visit states that are different relates to incentivizing the agent to explore states that are novel.
- The theoretical results seem a bit irrelevant. While it is nice to know the connection between the inverse dynamic bisimulation metric and the value difference / optimal value function, it remains elusive how this could be beneficial to exploration.


**Questions:**

- If the intuition behind bisimulation-based exploration is that the potential function corresponds to TD error, then how is it better than an exploration method which directly incentivizes the visitation of transitions with large TD error? Where does the improvement come from?
- How does the value difference bound in Theorem 2 relate to TD error? The TD error includes the current timestep reward and the discount factor, but the value difference does not.
- It seems that even at convergence, the bisimulation metric would still be large between adjacent but critically different states (e.g. states that have vastly different rewards). How does the method know when to stop exploring in this case?
- How does the method perform so well in the sparse reward setting, when the bisimulation metric explicitly includes a reward term?
- Can you include comparisons with the variant of the algorithm w/o inverse dynamics on Atari games?


**Limitations:**

The authors mention that their method may encounter limitations when tackling prolonged and hard exploration. But this is a rather vague statement. It would help to elaborate on specific settings that their method struggles with.

---

> ### Author Rebuttal · Authors · 2023-08-09
>
> Thank you for your insightful reviews. The requested exp have been included in the PDF file. For all the references mentioned in the response, please find the reference list in the global comment.
>
> *W1:.. the intuition of bisimulation metric for doing so remains largely unclear... However, it is unclear how incentivizing the agent to visit states that are different relates to incentivizing the agent to explore states that are novel...*
>
> **Response**: Recent studies, such as RIDE[A] and PBRS[B], have investigated the encouragement of agent exploration through state differences. However, these methods encounter issues of inefficiency and restricted scalability, compounded by their reliance on pre-existing knowledge (see lines 28-46 for reference). We endeavor to tackle these challenges in our work by introducing a perspective rooted in bisimulation metric-based state differences. A visual representation of this concept can be observed in Figure 2 (lines 158-184):  the agents can discover novel states by additionally considering the value difference, and it's acknowledged by reviewer dnFb (Strength point 3),  pk3V (Strength point 1) and  AtiU (Strength point 2). As shown in Figure 1, spikes in the curve (high state difference) correspond to pivotal moments of Mario's exploration, like jumping, boarding, and raising the flag. These actions lead to substantial state changes, including Mario reaching novel states like getting on the hoverboard. Conversely, in the 2nd and 3rd frames, we observe that Agent Mario can become stuck, with the state remaining almost unchanged during this interval.
>
> *Q1:If the intuition ... is that the potential function corresponds to TD error, how is it better than exploration method which directly incentivizes the visitation of transitions with large TD error? Where does the improvement come from?*
>
> **Response**: The summary of exploration methods can be found in Sec 2 and Table 3 in Appendix E. Even with the prioritization of visitation for transitions with large TD error, these methods fail to ensure *policy invariance* of the original MDP and lack *scalability* when compared to our approach (please refer to line 36-46 and line 60-69). Additionally, we employ Prioritized Experience Replay [C] to prioritize the visitation of transitions with large TD error in the most competitive baseline RIDE, the results can be found in Figure 2 in pdf of global comment, the performance of RIDE with PER declines or remains relatively unchanged across the six tasks. The reason behind is that RIDE only learns to explore state pairs with high TD error which significantly restrict the exploration. It's a trade-off between promoting the agent's exploration and visitation of transitions with large TD error for other exploration methods. Note that our approach takes into account the TD error between states derived from the metric. This means that we do not require prioritization of transitions with high TD error, as our model can autonomously assess its own performance. So we can achieve excellent exploration (see Figure 4 and 6) as well as accelerate convergence speed.
>
> *W2:theoretical results seem a bit irrelevant...it remains elusive how this could be beneficial to exploration*
>
> **Response**: The theoretical analysis is essential in supporting the evidence of our contribution, *policy invariance* and *more efficient exploration* (see line 60-66).  Thm 1 offers the evidence of the convergence of our metric. Thm 2 is the guarantee of how our method achieves **efficient** exploration by considering value difference  so that the training efficiency can be improved. Thm 3 and 4 analyze the relationship between our potential function and the optimal value function of the modified MDP and the original MDP.  Since Eq(10) is satisfied, the learning process of optimal value function can be more efficient by focusing on the non-zero V-values. Thank you and we will enhance this part in the revised version.
>
> *Q2: How does the value difference bound in Theorem 2 relate to TD error...*
>
> **Response**:The TD error is defined as: $\delta_t=R_{t+1}+\gamma V\left(S_{t+1}\right)-V\left(S_t\right)$. Intuitively, $\gamma$ is a constant during training, if the value difference of state $S_t$ and $S_{t+1}$ is large, the shaping reward $F$ evaluated by potential function will be large, which means $R = R^e +F$ will be large ($R^e$ is the external reward from the environment). So the TD error of the transition will be large if the value difference between adjacent states is large.
>
> *Q3: ..even at convergence, the bisimulation metric would be large between adjacent different states...How does the method know when to stop exploring in this case?*
>
> **Response**:When the policy is at convergence, please refer to sec 3 (line 122-126), our shaping reward is potential-based and our metric is proved to converge to a fixed point (see thm1), so it won't affect the optimal policy of the original MDP. As the policy converges to the optimal policy, the shaping reward will have no impact on training, so the agent *stops* exploring after the convergence of optimal policy, even in the case where states that have vastly different rewards.
>
> *Q4:How does the method perform so well in the sparse reward setting, when the bisimulation metric explicitly includes a reward term?*
>
> **Response**: Please refer to Eq(4), in sparse reward setting, the shaping reward mainly relies on the last two terms in Eq(4) which are the difference between transition distribution and action outcomes, so the agent will try to maximize the reward by exploring actions and transitions distributions that have large difference, agents are encouraged to take more diverse actions to maximize the last two terms in Eq(4) so that the exploration is effectively promoted.
>
> *Q5: Can you include comparisons with the variant of the algorithm w/o inverse dynamics on Atari games?*
>
> **Response**: Yes, we have included in Figure 1 in the pdf of the global comment.

---

> > ### Comment · Reviewer_naKC · 2023-08-16
> >
> > Thanks for addressing my comments and providing additional results. I am convinced that this work has a substantial contribution and will adjust my score accordingly.

---

> > > ### Author Response · Authors · 2023-08-17
> > >
> > > Thank you for acknowledging our contribution. We are confident that your valuable suggestions will undoubtedly enhance the quality of our paper.

---

### Author Rebuttal · Authors · 2023-08-09

We thank all reviewers for their valuable comments, and we summarize the major concerns regarding to the reviewers as follows:

### Sparse reward setting and more challenging environments
According to the review of the 1st reviewer naKC, there are questions about how our method can achieve good performance in sparse reward setting, and the 3rd reviewer pk3V suggests that we should evaluate our method on a more sparse reward goal-conditioned environment. The 4th reviewer AtiU suggests that we should also evaluate our method in more challenging exploration scenarios. To address this, we have included additional experiments in more challenging environments in the pdf and provide the analysis of how our method can perform well in the individual responses.

### Theorem connection and discussion
The 1st reviewer naKC has some concern how the theoretical analysis is beneficial to exploration, the 2nd reviewer dnFb has some questions on theorem 2, and the 3rd reviewer pk3V suggests that we should improve the the discussion about the necessity and importance of our theoretical results. lastly, the 4th reviewer AtiU has some concern that Theorem 1-3 closely resemble previous work. To address this, we had detailed the contribution and necessity of our theoretical analysis in each individual response.

### Intuition and questions of model-based approaches
The 1st reviewer has concern on the intuition behind our method, and the 2nd reviewer suggests that we should make comparison with model-based approaches. To address this, we had explained our intuition referring to Section 4, and we had clarified that our method is not model-based and detailed the comparison of model-based approaches in the individual responses.

We thanks again for all the reviewers putting time and care into reviewing the paper, and we had answered all the reviewer's questions and minor comments in the individual response. To facilitate cross-referencing, we present all the references utilized in the response here.

### Reference list
> [A] Raileanu R, Rocktäschel T. Ride: Rewarding Impact-Driven Exploration for Procedurally-Generated Environments. ICLR 2020
>
> [B] Ng A Y, Harada D, Russell S. Policy invariance under reward transformations: Theory and application to reward shaping. ICML 1999
>
>[C] Schaul T, Quan J, Antonoglou I, et al. Prioritized experience replay. arXiv 2015.
>
>[D] Kaiser L, Babaeizadeh M, Milos P, et al. Model-based reinforcement learning for atari. ICLR 2020.
>
>[E] Pathak D, Agrawal P, Efros A A, et al. Curiosity-driven exploration by self-supervised prediction. ICML 2017.
>
>[F] Badia A P, Sprechmann P, Vitvitskyi A, et al. Never give up: Learning directed exploration strategies. ICLR 2020.
>
>[G] Sheikh H, Phielipp M, Boloni L. Maximizing ensemble diversity in deep reinforcement learning. ICLR 2021.
>
>[H] Li C, Wang T, Wu C, et al. Celebrating diversity in shared multi-agent reinforcement learning. NIPS 2021.
>
>[I] Xu Z, van Hasselt H P, Silver D. Meta-gradient reinforcement learning. NIPS 2018.
>
>[J] Burda Y, Edwards H, Storkey A, et al. Exploration by random network distillation. arXiv 2018.
>
>[K] Rui Zhao, Yang Gao, Pieter Abbeel, Volker Tresp, Wei Xu: Mutual Information State Intrinsic Control. ICLR 2021.
>
>[L] Eysenbach B, Zhang T, Levine S, et al. Contrastive learning as goal-conditioned reinforcement learning. NIPS 2022.
>
>[M] Zheng Z, Oh J, Singh S. On learning intrinsic rewards for policy gradient methods. NIPS 2018.
>
>[N] Mazzaglia P, Catal O, Verbelen T, et al. Curiosity-driven exploration via latent bayesian surprise. AAAI 2022.

---

### Decision · Program_Chairs · 2023-09-21

**Decision:**

Accept (poster)

**Comment:**

This paper has been positively evaluated by all reviewers. The rebuttal phase has been very fruitful in providing further clarifications to some reviewers, which increased their initial scores. The additional clarifications about the theory and some additional results in the rebuttal have been crucial to reaching a positive agreement about this work.

I encourage the authors to address all the comments and to incorporate the recommended improvements in the final version.